# Mesenchymal Stromal Cell-Derived Extracellular Vesicles for Neonatal Lung Disease: Tiny Particles, Major Promise, Rigorous Requirements for Clinical Translation

**DOI:** 10.3390/cells11071176

**Published:** 2022-03-31

**Authors:** Flore Lesage, Bernard Thébaud

**Affiliations:** 1Sinclair Centre for Regenerative Medicine, Ottawa Hospital Research Institute, Ottawa, ON K1H 8L6, Canada; bthebaud@toh.ca; 2Children’s Hospital of Eastern Ontario Research Institute, Ottawa, ON K1H 8L6, Canada; 3Department of Cellular and Molecular Medicine, University of Ottawa, Ottawa, ON K1H 8L6, Canada; 4Division of Neonatology, Department of Pediatrics, Children’s Hospital of Eastern Ontario, Ottawa, ON K1H 8L6, Canada

**Keywords:** bronchopulmonary dysplasia, prematurity, lung disease, extracellular vesicles, exosomes, mesenchymal stromal cells, cell-based therapies, animal models

## Abstract

Extreme preterm birth disrupts late lung development and puts newborns at risk of developing chronic lung disease, known as bronchopulmonary dysplasia (BPD). BPD can be associated with life-long complications, and currently no effective treatment is available. Cell therapies are entering the clinics to curb complications of extreme preterm birth with several clinical trials testing the feasibility, safety and efficacy of mesenchymal stromal cells (MSCs). The therapeutic effect of MSCs is contained in their secretome, and nanosized membranous structures released by the MSCs, known as extracellular vesicles (EVs), have been shown to be the therapeutic vectors. Driven by this discovery, the efficacy of EV-based therapy is currently being explored in models of BPD. EVs derived from MSCs, contain a rich cargo of anti-inflammatory and pro-angiogenic molecules, making them suitable candidates to treat multifactorial diseases such as BPD. Here, we review the state-of-the-art of preclinical studies involving MSC-derived EVs in models of BPD and highlight technical and regulatory challenges that need to be addressed before clinical translation. In addition, we aim at increasing awareness regarding the importance of rigorous reporting of experimental details of EV experiments and to increase the outreach of the current established guidelines amongst researchers in the BPD field.

## 1. A Brief Description of Bronchopulmonary Dysplasia

Late lung development is a crucial period of lung formation and maturation, which aims at increasing the surface available for gas exchange. This stage of lung development occurs from 36 weeks of gestation and is suggested to last until the eighth year of life [1]. During this period the most distal respiratory units are being formed, the so-called alveoli. Within these units, respiration occurs across a thin epithelial/microvascular barrier. Preterm birth disrupts late lung development and puts newborns at risk of developing a chronic lung disease known as bronchopulmonary dysplasia (BPD). BPD is characterized by a histological pattern of “alveolar simplification” (fewer and larger alveoli) and decreased microvascular density. The large airways show mild injury and mild fibrosis is present. Small airway injury is increasingly described in human and experimental BPD, consistent with long-term impaired respiratory dysfunction [2]. Over the past several decades, advances in perinatal care, including the implementation of gentler ventilation techniques combined with tailored administration of oxygen, surfactant therapy and antenatal corticosteroid administration, have increased survival of even the most extremely low gestational age neonates born at 23 weeks of gestation. However, BPD persists with an incidence of nearly 45% for infants born before 29 weeks of gestation [3,4,5]. BPD remains the most common chronic disease in children and a leading cause of death [6]. Survivors suffer from life-long respiratory complications, with an increased incidence of asthma, early-onset emphysema and pulmonary vascular disease, and are at greater risk of recurrent hospitalizations and neurodevelopmental delay [7,8,9,10]. There is currently no treatment available that improves the complicated pathogenesis of BPD patients.

## 2. The Rationale for Using Mesenchymal Stromal Cells as Immune Modulating Agents in Patients with BPD

Stem cells, first described by Till and McCulloch in 1963, are undifferentiated cells that have two important properties: (1) self-renewal and (2) the ability to differentiate into at least one other cell type. Stem cells are widely investigated in the context of several diseases, including for the regeneration of the preterm lung affected by BPD. Today, more than 60 years after the first successful bone marrow transplant, hematopoietic stem cell transplantation is a routine procedure in the treatment of certain types of cancer, and blood and immune system disorders that affect the bone marrow. However, the clinical use of other than hematopoietic stem and progenitor cells for regenerative purposes is still limited and focuses mainly on mesenchymal stromal cells (MSCs). MSCs were for the first time described in the 1970s as fibroblasts derived from the bone marrow with colony-forming capacity [11,12]. The exponential increase in publications involving MSCs around the turn of the century, prompted the International Society for Cellular Therapy to define minimal criteria for these cells, including (1) plastic-adherence, (2) expression of cluster of differentiation (CD) 105, CD73 and CD90, and lacking the expression of CD45, CD34, CD14 or CD11b, CD79alpha or CD19 and human leukocyte antigen isotype DR (HLA-DR) surface molecules, and (3) ability to differentiate into osteoblasts, adipocytes and chondroblasts in vitro [13]. MSC can be derived from a variety of tissues and various in vitro and in vivo studies show that MSC can modulate inflammation, improve organ function and prolong survival (Figure 1). The underlying mechanisms of MSC-mediated immune modulation are not yet completely understood and depend on a variety of influencing factors (including tissue source, culture conditions and in vivo environment), however, certain pathways have been repeatedly demonstrated. MSCs induce functional changes in macrophages, dendritic cells, T cells, B cells and natural killer cells and these changes are tailored to the specific microenvironment that the MSCs encounter in vivo [14,15,16,17,18,19]. In particular, the MSC-induced anti-inflammatory effects of macrophages and regulatory T cells play a critical role in their immunomodulatory capacity. Because of the ease of isolating and expanding MSCs, as well as their beneficial therapeutic effects, MSCs have acquired widespread interest with a multitude of (pre)clinical trials investigating MSC intervention across a variety of diseases involving regeneration and immunomodulation. In the context of BPD, umbilical cord (UmC) derived MSCs have proven to protect lung development in neonatal rats exposed to high concentrations of oxygen, which induces a phenotype reminiscent of BPD [20]. The therapeutic benefit of MSCs to treat experimental BPD was confirmed in systematic reviews and meta-analyses [21,22]. Early phase clinical trials are now underway to test the feasibility, safety and efficacy of MSCs in infants at risk of developing BPD [23,24,25,26].

## 3. Cell Therapy without the Cell: The Promise of Extracellular Vesicles

In recent years, MSC research has shifted from not only testing cell products, but also testing cell-derived products such as extracellular vesicles (EVs), which may represent a more practical product for therapy than intact cells. The ability of MSC to communicate with and influence the environment has been attributed to cell contact-dependent mechanisms, but most interestingly also to the secretion of paracrine factors such as bioactive compounds and EVs. Indeed, administration of cell-free MSC-conditioned medium results in lung protection in experimental BPD [20]. EVs comprise a heterogeneous population of nanosized vesicles delimited by a lipid bilayer that do not contain a nucleus and that are released by every cell type in the body (Figure 2). EVs contain bioactive agents that have an essential role in intercellular communication, both during organ homeostasis and pathology-related changes. The mechanisms by which EVs mediate cell-to-cell signaling are only partly understood. Upon absorption from the environment, the EVs’ cargo has the potential to influence various cellular processes, including gene transcription, antigen presentation, cell proliferation, differentiation or apoptosis, and others [27,28].

EVs are categorized into two main subpopulations based on their physical characteristics: small and large EVs [29]. Small EVs, also referred to as exosomes, are 30–100 nm in size and are secreted through the endosomal pathway, involving the fusion of multivesicular endosomes with the cell membrane. Large EVs, also referred to as microvesicles or microparticles, range in size from 50 nm to a few micrometers and are generated by shedding from the plasma membrane, having a membrane composition similar to that of the cell membrane. During their formation, bioactive compounds from the parent cell are packaged inside both small and large EVs, including cell-type specific protein combinations (enzymes, growth factors, receptors and cytokines) as well as nucleotides, lipids and metabolites. Exposing parent cells to varying culture conditions, including low pH or low oxygen concentrations, will influence the secretion and cargo of EVs [30,31].

The research into EV-based therapeutics has been fueled by their specific advantages over cellular approaches. Firstly, EVs hold a lower risk of tumor formation, autoimmune responses and toxic effects compared with cell therapy. Secondly, production, sterilization and storage are less complex than for cell-based products. Thirdly, EVs are biologically inert and their biological properties will not be influenced by the in vivo environment. Altogether, these EV-specific properties are believed to enable the generation of a robust, well-defined, ready-to-use, clinical-grade product.

## 4. Current State-of-the-Art for EV-Based Therapy in BPD Animal Models

Currently, 15 different articles explored the effects of MSC-EVs in newborn rodents with BPD (Table 1). Nearly all BPD animal models were established by housing rodents under hyperoxia conditions, ranging from 60–95% O_2_. The team of Dr. Kourembanas was the first to show that MSC-EVs improved pulmonary parenchymal and vascular development in newborn mice exposed to hyperoxia. They reported on both the preventative capacity of UmC-MSC-EVs when administered early, after the onset of hyperoxia-induced injury [32], as well as on their capacity to reverse established injury when administered later [33]. EVs improved alveolar simplification, pulmonary fibrosis, vascular remodeling, blood vessel loss, right ventricular hypertrophy and lung function. Since those initial reports, Dr. Kourembanas’ team has published a substantial amount of data elucidating the mechanistic pathways behind the MSC-EVs’ observed therapeutic effect [32,33,34,35]. MSC-EV treatment restored normal levels of pulmonary myeloid cells disrupted by hyperoxia-induced lung injury and direct interaction of MSC-EVs with these myeloid cells promoted an immunosuppressive, CCR2-associated myeloid cell phenotype. In addition, MSC-EV-educated bone marrow-derived myeloid cells also showed a therapeutic effect in BPD mice. Functional assays demonstrated that these therapeutic effects were driven by phenotypically and epigenetically reprogrammed monocytes, which induced further modulation of the CCR2/CCL2 axis and the recruitment of monocyte-derived suppressive cells expressing high levels of the anti-inflammatory Arginase 1 (Arg1) and Interleukin (IL) −10. Notably, this study showed the biodistribution of EVs after intravenous delivery. The EVs migrated from the injection site to the lungs and the liver already at 5 min after injection and over the next 24 h, the signal in the lungs slightly faded while it increased in intensity in the liver. Hyperoxia exposure also has an effect on organs outside of the cardio-pulmonary compartment, including the thymus gland, which is crucial for the neonatal immune system and the development of adaptive immunity. Reis et al. demonstrated that hyperoxia induced an involution of the thymic medulla and associated this lesion with the disrupted generation of FoxP3+ regulatory T cells and T cell autoreactivity [34]. The treatment with MSC-EVs restored thymic architecture and thymocyte functionality. Furthermore, by using single cell RNA-sequencing (scRNA-seq) to unravel the transcriptome of the thymic tissue, they established that MSC-EVs exerted their beneficial effects through the thymic medullary antigen presentation axis, by enriching antigen presentation and antioxidative-stress related genes in dendritic cells and medullary epithelial cells.

Other reports highlight the importance of different pathways through which the therapeutic actions of MSC-EV might be mediated in BPD animals. For instance, Chaubey et al. attributes the effects of UmC-MSC-derived EVs to their enrichment in tumor necrosis factor-stimulated gene-6 (TSG-6) [39]. TSG-6 levels typically increase when MSC are cultured in serum-free conditions, which are the conditions that are used to produce conditioned medium for EV isolation. TSG-6 is a key mediator in the immunosuppressive properties of MSC. It is produced in response to the release of inflammatory factors tumor necrosis factor α (TNF-α) and IL-1β, and it induces a phenotypic shift in macrophages from a proinflammatory M1 profile towards an anti-inflammatory M2 profile. When combining a TSG-6 neutralizing antibody with MSC-EV treatment in BPD animals, or when treating with EVs derived from UmC-MSC that were transfected with a TSG-6 siRNA, only very limited beneficial effects could be observed as compared to MSC-EV treatment alone.

Consistent with these anti-inflammatory properties, Abele et al. further elucidated the therapeutic potential of bone marrow-derived MSC-EVs in a BPD model involving chorioamnionitis, which is a major risk factor for developing BPD. Antenatal delivery of an endotoxin to fetal rats induced intrauterine inflammation and perturbed both placental vascular development and postnatal distal lung growth, all of which was reversed by antenatal MSC-EV treatment [41]. In addition, Lithopoulos et al. demonstrated the immune modulating properties of MSC-derived EVs in a neonatal lung injury model involving the injection of an endotoxin, followed by 8 h of ventilation (40% O_2_) [45]. The lungs of EV-treated animals showed decreased expression of pro-inflammatory markers (IP-10) and increased expression of anti-inflammatory markers (IL-4, IL-13). Ventilated mice treated with MSC-EVs showed significant improvement in lung architecture and vessel formation. Interestingly, MSC-EV treatment also rescued in vitro neural progenitor cell function, suggesting a dual lung and brain protective effect.

EVs are vehicles of a variety of bioactive molecules, including microRNAs (miRNAs). miRNAs are small non-coding RNAs that modulate gene expression at the post-transcriptional level and are a crucial regulator of lung development. miR-21-5p expression is reduced in rats suffering from hyperoxia-induced lung injury [47]. Adipose tissue (AT) -derived MSC-EVs have been shown to attenuate the effects of hyperoxia on neonatal mice by transferring miR-21-5p into lung cells [44]. Ultimately, this resulted in the modulation of C/EBPα expression, which is a critical transcription factor for perinatal lung maturation and is known to be involved in lung epithelial repair post injury [48].

Ai et al. reported on the modulation of WNT5a by UmC-MSC-EV therapy in BPD rats [46]. WNT5a is a crucial mediator of the transdifferentiation of alveolar type 2 (AT2) cells into type 1 cells, which is increased in hyperoxia-induced neonatal lung injury [49]. UmC-MSC-EVs are able to delay in vitro AT2 cell transdifferentiation, but further research is required to validate these findings in vivo. 

Another highly studied pathway in BPD therapeutics is the vascular endothelial growth factor (VEGF) pathway. It was previously reported that VEGF knockdown attenuates the in vivo protective effects of MSC therapy on alveolarization, angiogenesis and inflammation in lung injury models [50]. Likewise, EVs derived from MSC with a small interfering RNA induced knockdown of VEGF render diminished therapeutic effects in BPD rats as compared to EVs derived from control MSC, suggesting that part of the therapeutic effects of MSC and their EVs is mediated via VEGF-dependent pathways [38]. Interestingly, they report that (VEGF-containing) MSC-EVs colocalize most frequently with pericytes and alveolar macrophages within the lung tissue, and not with vascular cells. These findings suggest that the effects of MSC-EVs on angiogenesis are mediated by EV engulfment into pericytes and subsequent paracrine crosstalk between pericytes and endothelial cells. Indeed, it has been reported that pericytes potentiate endothelial cell survival and the stability of microvessels by VEGF-A expression [51]. Finally, also You et al. confirmed the role of VEGF-A and the upstream PTEN/Akt signaling pathway (known to regulate cellular proliferation, apoptosis and angiogenesis [52]) mediating the lung protective effect of UmC-derived MSC-EVs [43].

Lastly, Porzionato et al. reported the therapeutic effect of repeated injections with UmC-derived MSC-EVs in BPD rats, both in the short- and long-term [36,42]. Interestingly, they developed a clinical-grade protocol to produce the MSC-EVs, including rigorous quality control checks during MSC culture and the generation of conditioned medium (cell viability, cell growth rate, continuous monitoring of cell culture parameters, etc).

## 5. EV-Based Therapy in BPD Animal Models: Additional Research Required before Clinical Translation

Apart from UmC-derived MSC-EVs, EVs derived from MSC from other tissue sources are understudied in the context of BPD. EVs derived from MSC isolated from the bone marrow and the amniotic membrane were shown to have a protective effect on alveolar and lung vascular development in hyperoxia-induced lung injury in neonatal rats, but additional research is needed to understand the potential of MSC-EV derived from these alternative tissue sources and to unravel which source is superior [37,40,41]. A meta-analysis including all studies where EVs derived from MSC from different tissue sources were used as a treatment in a variety of experimental models of lung injury (including BPD), indicated no difference between tissue sources in regards to the EVs’ therapeutic efficiency [53]. The lack of head-to-head comparisons of EVs derived from MSC from different tissue sources is also seen when looking at the MSC as a therapeutic product themselves. Very little is known about MSCs derived from alternative tissue sources, with bone marrow-derived MSC still being the most studied, both in terms of in vitro and vivo characteristics. Side-by-side comparisons of MSC derived from different tissues are very limited and the only comparison in a BPD model, was between MSC derived from the UmC and from the AT [54]. Both UmC-MSC and AT-MSC significantly improved alveolarization. However, a superior effect of UmC-MSC on modulating angiogenesis, lung inflammation and cell death was observed. UmC-MSC produced higher levels of VEGF and hepatocyte growth factor in vitro, which might explain their elevated in vivo potency as compared to adipose tissue-derived MSC. Furthermore, within the context of BPD, a meta-analysis on all in vivo studies involving MSC, reported a significant effect of MSC therapy on alveolarization regardless of tissue source [22]. Interestingly, of all included studies, 32% used UmC-MSC and 68% used bone marrow-derived MSC. In line with the need to compare the therapeutic potential of different MSC tissue sources, we also identify the need to compare the efficacy of cell versus EV therapy. Three studies, conducted a direct comparison between cells and EV in the context of BPD and–generally–did not identify any major difference [38,39,40]. In terms of alveolarization, Porzionato et al. only identified a significant effect of EVs, whereas Li et al. only identified a significant effect of cells. However, when also considering secondary outcomes such as lung inflammation and lung angiogenesis both cell and EV therapy had a significant effect. Differences in efficacy of cell versus EV therapy should be interpreted in the context of the given dosage, which is in most cases established empirically, and with studies comparing different dosages still missing. The beneficial effects of cell therapy are most likely not solely mediated by EVs, but also by the release of proteins and nucleotides. In addition, EVs released from in vivo transplanted MSC are most likely different from EVs released from in vitro cultured cells in terms of quantity and cargo. So, when performing therapeutic experiments and when conducting side-by-side comparisons, different dosages have to be considered for both cell therapy and EV therapy, and these types of dose escalation studies where optimal therapeutic dosage is established in experimental BPD models are still missing.

## 6. Influencing the Bioactive Properties of EVs

The loading of cargo into EVs is currently being explored in order to enhance their effects for specific therapeutic applications. Two different approaches can be distinguished to engineer these drug delivery vehicles: exogenous loading (by incorporating small molecules, proteins or RNAs into isolated EVs) and endogenous loading (by enabling the parent cells to incorporate specific bioactive compounds into EVs during biogenesis). Exogenous therapeutic cargo is loaded into isolated EVs by co-incubation, electroporation or sonication. Endogenous loading is achieved by genetic modifications of the parent cell aiming at overexpressing a specific RNA or protein, which is then incorporated into the secreted EVs [55].

Bioengineered MSC-derived EVs have not been explored yet in the context of experimental BPD. Some interesting candidate MSCs are on the horizon, including good manufacturing practice manufactured Interferon γ-primed MSC with heightened intrinsic anti-inflammatory capacity [56]. Another interesting approach is to educate the parent MSC in vitro with the inflammatory environment the EVs will encounter in vivo. Priming MSC with the secretome of activated microglia resulted in the release of EVs with enhanced immune regulatory potential to fight neuroinflammation [57]. Similarly, MSCs educated in vitro by inflammatory or hyperoxia-exposed lung cells could yield EVs with enhanced therapeutic efficacy in the context of BPD.

## 7. Standardization of Isolation and Characterization Methods for EV Research

Much of the variability within the EV field originates from the absence of standard isolation and characterization methods [29]. Obtaining pure EV populations is one of the greatest challenges when studying EVs. There is an overlap of biophysical properties between the different EV subpopulations and most parent sources (biological fluids or conditioned culture medium) contain co-isolating contaminants. EV isolation techniques can be divided into five groups, each exploiting a particular biophysical EV property: (i) ultracentrifugation (UC)-based techniques; (ii) precipitation-based techniques; (iii) size-based techniques; (iv) immunoaffinity capture-based techniques; and (v) microfluidic-based techniques [58,59]. The EV isolation method used is known to have a significant effect on EV content and quality. Table 2 summarizes, for each MSC-derived EV-based preclinical study in the BPD field, the isolation technique used. Eight studies used UC. The most commonly used UC-technique in the EV field is differential centrifugation which pellets different cellular structures based on their sedimentation rate by sequentially increasing the relative centrifugal force (RCF), with large EVs pelleting at an RCF of 20,000 g and small EVs at an RCF of 100,000 g [60,61]. However, it has to be noted that three studies did use UC without gradually increasing the RCF, resulting in a heterogenous product containing both large and small EVs together [38,40,43]. Both studies of Porzionato employed the size-based concentration technique tangential flow filtration (TFF), while the studies from the Kourembanas’ group typically employ a combination of both TFF and UC using a clinical-grade density gradient, resulting in a highly purified product. In order to promote standardization in the characterization of EV preparations and to overcome the lack in complete reporting, The International Society for EVs (ISEV) updated their Minimal Information for Studies of EVs (MISEV) guidelines in 2018 [29]. The aim of these guidelines is to enable the interpretation and comparison of the results of different studies and to reach general conclusions. The first MISEV requirement includes a quantification of the source of the EVs, both in terms of starting amount (e.g., number of cultured cells) and of the isolated EVs (e.g., protein concentration or particle count, as quantified by nanoparticle tracking analysis (NTA)). Secondly, the protein composition of EVs has to be analyzed and the presence of at least three EV-associated proteins, including both transmembrane (e.g., CD9, CD81) and cytosolic proteins (e.g., TSG101, ALIX), and the absence of proteins present in non-EV co-isolated structures (e.g., GM130, CALNEXIN) has to be shown. Thirdly, MISEV requires the characterization of single vesicles by at least two methods, such as electron microscopy and NTA.

Table 2 summarizes all EV-based studies in BPD animal models and their fulfillment of MISEV criteria. Out of fifteen studies, only five studies fulfilled all three MISEV requirements [33,34,35,39]. Most studies did not meet the second requirement to characterize the EV protein content (eight out of fifteen). While most studies did characterize the presence of transmembrane proteins such as tetraspanins, they did not characterize the presence of cytosolic EV proteins or the absence of non-EV proteins present in co-isolated structures. The third requirement, to confirm the presence of single EVs, was only confirmed in eleven out of fifteen studies, whereas the first requirement, to quantify the EV content, was met by all studies.

In order to further facilitate standardization of EV research and to increase systematic reporting on EV biology and methodology, as well as to create awareness and increase outreach regarding these topics, the EV-TRACK knowledgebase was developed (evtrack.org) [62,63]. This community-driven database invites authors to submit details on EV isolation and characterization via an online template composed of nine experimental parameters which are considered indispensable for unambiguous interpretation and replication of EV experiments. By centralizing methodological variables, EV-TRACK aims to keep track of the current state-of-the-art. In addition, data miners can use the EV-TRACK database to identify reporting or characterization deficits within the field. ISEV strongly encourages authors to submit experimental details to EV-TRACK. None of the identified articles in this review was submitted to EV-TRACK.

In conclusion, EV-based studies within the field of BPD are currently hard to interpret and to compare with one another as different isolation methods enrich for EVs with diverse composition and variable purity, and as information of the EV content and biophysical properties is generally missing or incomplete. With this review, we hope to increase awareness regarding the importance of rigorous reporting of experimental details of EV experiments and to increase the outreach of the MISEV initiative amongst researchers in the BPD field.

## 8. Future Directions for the EV Field

EV research is impacted by several technical challenges which need to be addressed within the coming years. Firstly, we identify the need for improved isolation methods yielding more homogeneous EV preparations in less time and at a lower cost. Nowadays, the isolation technique used is mostly determined arbitrarily, based on the equipment available to the researchers, the source material, the downstream application and the required homogeneity. Secondly, we need better and more sensitive tools to characterize EVs. EV characterization is often limited by the relative paucity of the material as the most common methods (e.g., western blot) require a lot of EVs. The next MISEV guidelines are currently being established and will aim at aiding with these two technical shortcomings by (1) educating/advising researchers on the pros and cons of different EV isolation methods and (2) facilitating EV characterization with the established methods by creating a panel of protein and non-protein markers to use for different EV subtypes, including reliable antibodies for different applications [64].

Apart from solving technical challenges, there is also a need for a deeper understanding of the mechanisms by which EVs function. Basic EV biology still must be elucidated, including the EV cargo, from mRNAs to proteins and lipids, the sorting/packaging process of specific cargo into cells with a specific physiology, the selection process of recipient cells and the delivery mechanism of cargo to these cells. Specialized community-driven databases such as EVpedia, Vesiclepedia, Exocarta and the more recent ExoRbase and EVmiRNA aim at ensuring public access to cargo information and link these data with technical specifications in order to enable cargo association with specific EV subsets [65,66,67,68,69]. EVs are gaining increased attention for the transfer of mRNA and miRNA, with the parent cell having the ability to purposefully package specific RNA content into EVs. Various RNA-binding proteins as well as membrane proteins are being identified in assisting with the sorting process into EVs [70]. The sorting of protein cargo into EVs is fairly well understood, whereas not much is known about lipid sorting [71]. Remaining biological questions when thinking about employing EVs as a therapeutic product include the molecular signature of therapeutic EVs, as well as the earlier raised questions about dose versus response and about the efficacy of a cell product versus an EV product. Furthermore, various technologies now enable the engineering of EVs, which holds promise for enriching EVs with the therapeutic cargo or for targeting EVs for functional use instead of for lysosomal degradation. However, more research is required to pinpoint the molecules involved in these processes.

Lastly, before EV therapy can be translated for clinical applications, there is a need for a regulatory framework. This includes guidelines regarding current good manufacturing practices (cGMP), compliant production, quality control criteria, sterilization methods and standard operating procedures for reproducibility. The ISEV published a position paper providing important considerations regarding safety and efficacy, clinical-grade manufacturing and regulatory issues, and plans on implementing further guidelines in the next MISEV [64,72].

In conclusion, since their initial discovery 30 years ago, EVs derived from MSC have gained great attention for different clinical applications involving tissue regeneration and immune modulation. The efficacy of MSC-derived EVs in experimental BPD has been shown robustly by several research teams in the past 4 years. In order to allow clinical translation, some technical and biological advancements are needed. To generate a large-scale clinical-grade product, more efficient EV isolation and characterization techniques are required. For safeguarding safety and efficacy, biological questions regarding the EV cargo, action mechanism, pharmacokinetics and biodistribution, as well as administration and dosage need to be answered. In addition, a regulatory framework will have to be set in place.

## Figures and Tables

**Figure 1 cells-11-01176-f001:**
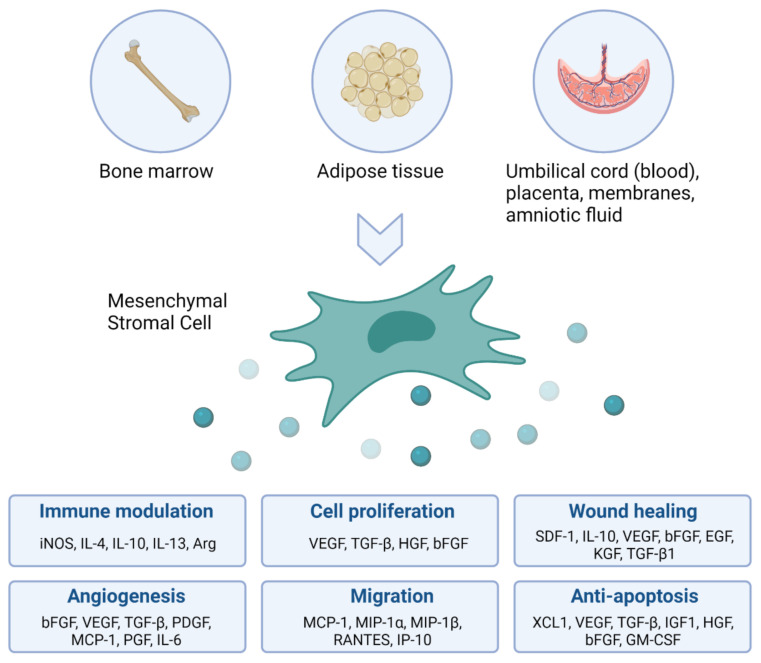
The therapeutic function of mesenchymal stromal cells (MSCs). MSCs can be isolated from a variety of tissue sources. Cultured MSCs have been used for several therapeutic purposes, both in experimental research and clinical applications. MSCs have the ability to modify their pleiotropic effects based on the in vivo environment they encounter. The secretome of MSCs is known to be involved in immune modulation, cell proliferation, wound healing, angiogenesis, migration and anti-apoptosis. Figure created with Biorender.com, accessed on 28 March 2022.

**Figure 2 cells-11-01176-f002:**
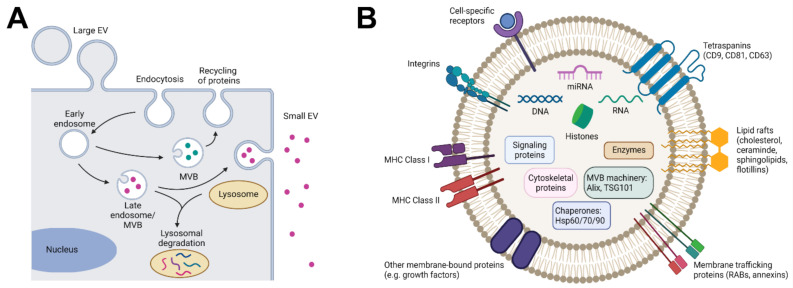
Schematic representation of the formation and composition of extracellular vesicles (EVs). (**A**) Large EVs are generated by budding from the plasma membrane. Small EVs originate in the endocytic pathway by the formation of an early endosome via endocytosis at the plasma membrane. Intraluminal vesicles are formed by the inward budding of the membrane of the late endosome or multivesicular body (MVB). MVBs are targeted for degradation by the lysosome or for secretion by fusion with the plasma membrane. This results in the release of small EVs in the extracellular space. (**B**) The membrane of small EVs contains proteins acquired during their biogenesis (G-proteins (Rabs), Flotillin-1 (FLOT1), etc.), as well as proteins involved in targeting recipient cells (integrins and tetraspanins (CD9, CD81, CD63)) and immune modulation (MHC Class I and MHC Class II receptors). The cytosolic cargo of small EVs includes cell-specific nucleic acids (DNAs, RNAs, miRNAs), histones (H4), proteins from the MVB machinery (ALIX, TSG101), chaperones (HSP70, HSP90, HSP60), signaling proteins (HIF-1α, β-catenin), enzymes (GAPDH, pyruvate) and cytoskeletal proteins (actin, tubulin). Figure created with Biorender.com, accessed on 28 March 2022.

**Table 1 cells-11-01176-t001:** In vivo experimental details of current MSC-derived EV-based preclinical studies in models of bronchopulmonary dysplasia.

Publication	Model	Species	EV Source	Dose Analysis	Dose	Route	Timing of EV Administration: Prevention	Timing of Data Collection
Willis 2018 [32]	Hyperoxia (75% O_2_, PN1-7)	Mouse	h UmC-MSC, hBM-MSC	Cell equivalent	0.5 × 10^6^	IV	PN4	PN7, PN14, PN42
Porzionato 2019 [36]	Hyperoxia (60% O_2_, PN1-14)	Rat	h UmC-MSC	Particle count	8 × 10^8^ at PN3; 4.5 × 10^8^ at PN7; 3 × 10^8^ at PN10	IT	PN3, PN7, PN10	PN14
Braun 2018 [37]	Hyperoxia (85% O_2_, PN1-14)	Rat	r BM-MSC	Protein concentration/particle count	15 µg/3.4 × 10^9^	IP	PN1—14, daily	PN14, PN21, PN56
Ahn 2018 [38]	Hyperoxia (90% O_2_, PN1-14)	Rat	h UCB-MSC	Protein concentration	20 µg	IT	PN5	PN14
Chaubey 2018 [39]	Hyperoxia (95% O_2_, PN1-4)	Mouse	h UmC-MSC	Cell equivalent	0.7 × 10^6^	IP	PN2, PN4	PN14
Li 2020 [40]	Hyperoxia (80% O2, PN1-14)	Rat	h AT-MSC	Protein concentration	300 ng	IT	PN7	PN14
Willis 2020 [33]	Hyperoxia (75% O2, PN1-14)	Mouse	h UmC-MSC	Cell equivalent	Early: 0.5 × 10^6^; Bolus late 1 × 10^6^; Serial late: 1 × 10^6^	IV	Early: PN4; Bolus late: PN18, Serial late: PN18-25-32-39	Early: PN60; Bolus late: PN28; Serial late: PN60
Abele 2021 [41]	Chorioamnionitis (Endotoxin, E20)	Rat	h BM-MSC	Cell equivalent	0.25 × 10^6^	IA	E20	PN14
Porzionato 2021 [42]	Hyperoxia (60% O_2_, PN0-14)	Rat	h UmC-MSC	Particle count	8 × 10^8^ at PN3; 4.5 × 10^8^ at PN7; 3 × 10^8^ at PN10	IT	PN3, PN7, PN10 and PN21	PN42
Reis 2021 [34]	Hyperoxia (75% O_2_, PN1-7)	Mouse	h UmC-MSC	Cell equivalent	0.5 × 10^6^	IV	PN4	PN14
Willis 2021 [35]	Hyperoxia (75% O_2_, PN1-14)	Mouse	h UmC-MSC	Cell equivalent	0.5 × 10^6^	IV	PN4	PN28
You 2021 [43]	Hyperoxia (85% O_2_, PN0-14)	Rat	h UmC-MSC	Protein concentration	20 µg	IT	PN7	PN14
Wu 2021 [44]	Hyperoxia (95% O_2_, PN1-3)	Mouse	m AT-MSC	Protein concentration	30 or 300 ng	IT	PN1	PN3
Lithopoulos 2022 [45]	Endotoxin (PN7/8) + Ventilation (PN9/10, 40% O_2_, 8 h)	Mouse	h UmC-MSC	Protein concentration/Particle count	0.005 μg/g; approximately 1 × 10^6^ particles/g	IT	PN9/10	8 h after EV delivery
Ai 2022 [46]	Hyperoxia (75%, O_2_, PN1-14)	Rat	h UmC-MSC	Protein concentration	10 or 15 µg	IP	PN4	PN14, PN21, PN42

Abbreviations: AT: Adipose tissue-derived; BM: bone marrow, BPD: bronchopulmonary dysplasia, EV: extracellular vesicles; h: human; IA: intra-amniotically; IT: intratracheal; IV: intravenously; MSC: mesenchymal stromal cells, PN: postnatal day, r: rat; UmC: umbilical cord.

**Table 2 cells-11-01176-t002:** EV isolation and characterization methods of current EV-based preclinical studies in models of bronchopulmonary dysplasia.

		REQ1: EV Quantification		REQ2: Protein Characterization			REQ3: Single EVs	
Publication	Isolation Method	BCA	NTA	Presence of Trans-Membrane Proteins	Presence of Cytosolic Proteins	Absence of non-EV Proteins	TEM	NTA
Willis 2018 [32]	TFF + UC		✓	CD63, CD9, CD81	HSP70		✓	✓
Porzionato 2019 [36]	TFF		✓	CD63, CD9, CD81	ANNEXIN V	ALBUMIN		✓
Braun 2018 [37]	UC	✓	✓	CD63, CD9, CD81			✓	✓
Ahn 2018 [38]	UC	✓	✓	CD63, CD9		GM130, FIBRILLARIN	✓	✓
Chaubey 2018 [39]	UC	✓	✓	CD63, CD81	ALIX1	TGN48	✓	✓
Li 2020 [40]	UC	✓		CD63, CD9, CD81	HSP70		✓	
Willis 2020 [33]	TFF + UC		✓	CD63, CD9, CD81	FLOT1, ALIX, TSG101	GM130	✓	✓
Abele 2021 [41]	TFF + UC	✓	✓					✓
Porzionato 2021 [42]	TFF		✓	CD63, CD9, CD81	ANNEXIN V	ALBUMIN		✓
Reis 2021 [34]	TFF + UC		✓	CD63, CD81	TSG101, SDCBP	CALNEXIN	✓	✓
Willis 2021 [35]	TFF + UC		✓	CD63, CD9	FLOT1, ALIX, TSG101	GM130, CALNEXIN	✓	✓
You 2020 [43]	UC	✓	✓	CD63	ALIX		✓	✓
Wu 2021 [44]	UC	✓	✓	CD63, CD9		CALNEXIN	✓	✓
Lithopoulos 2022 [45]	UC	✓	✓	CD63	FLOT1	CALNEXIN	✓	✓
Ai 2021 [46]	UC	✓	✓	CD63, CD9	FLOT1		✓	✓

Studies highlighted in light green partially fulfilled the MISEV2018 criteria for EV characterization, whereas studies highlighted in dark green fulfilled all MISEV2018 criteria. Abbreviations: BCA: bicinchoninic acid protein assay; NTA: nanoparticle tracking analysis; TEM: transmission electron microscopy; TFF: tangential flow filtration; UC: ultracentrifugation.

## Data Availability

Not applicable.

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
