# Peer review of "Mesenchymal Stromal Cell-Derived Extracellular Vesicles for Neonatal Lung Disease: Tiny Particles, Major Promise, Rigorous Requirements for Clinical Translation"

_cells, 2022, doi:10.3390/cells11071176_

Round 1
Reviewer 1 Report
The review has well described and exhaustive but I suggest adding a figure illustrating the physiopathology of the Bronchopulmonary dysplasia and the rationale for using MSCs as possible therapeutic approach.
I suggest to review the text because many words are syllabized in the text such us histo-logical , admin-istration,....
Author Response
Dear reviewer,
We greatly appreciate the time you have taken to review our manuscript and we thank you for the constructive feedback that helped improve our manuscript. Upon your request, we have included a figure on the therapeutic benefits of mesenchymal stromal cells and the rationale for using them in clinical applications, including bronchopulmonary dysplasia. In addition, we screened the text for mistakes against the English language.
Sincerely,
Dr. Bernard Thébaud
Dr. Flore Lesage
Reviewer 2 Report
The authors review the state-of-the-art of preclinical studies involving EV based therapy in models of bronchopulmonary dysplasia (BDP) and highlight technical and regulatory challenges that need to be addressed before clinical translation. They also point out the importance of rigorous reporting of experimental details of EV experiments. The authors found 12 different articles exploring the effects of MSC-EVs in newborn rodents with BDP. They conclude that the efficacy of MSC-EVs in experimental BPD has been demonstrated by several groups but their application in clinical trials requires technical and biological advancements.
This is a very extensive review and discussion of the literature concerning the efficacy of EV-based therapy in animal models of BPD. Overall the manuscript is well organized and well written, and I think it will be helpful for readers of Cells. Tables are clear and well structured.
The authors can consider following suggestions to improve the review:
In the paragraph “Current state-of-the-art for EV-based therapy in BPD animal models”, the authors should add these references and should add comments:
Bellio et al, Cytotherapy 2021 (Use of amniotic fluid as natural source of EVs in an experimental BPD model)
Lithopoulos MA et al, Am J Respir Crit Care Med 2022 (Use of MSC-EVs in a multifactorial lung injury model)
Danyang Ai et al, Stem Cells Dev 2022 (MSC-EVs suppress the transdifferentiation of rat alveolar type 2 epithelial cells via doxnregulation of WNT5a)
Wu Y et al, Stem Cell Rev Rep 2021 (Effect of Adipose Tissue derived MSC-EVs carrying miR-21-5p into lung cells)
Concerning the factors influencing the bioactive properties of EVs, the authors should also explain the possibility to modify EVs in order to enhance their efficacy. Different stimuli such as inflammation, hypoxia and engagement of Toll-like receptors are able to increase the production rate of EVs by MSCs as well as to increase their functions by modulation of their cargo (Guess AJ et al, Stem Cells Transl Med 2017; Markoutsa E et al, Mol Ther 2022). Ex vivo strategies to enhance EV efficacy should be also explained in this review.
There are some typographical errors in the text and tables.
Author Response
Dear reviewer,
We greatly appreciate the time you have taken to review our manuscript and we thank you for the constructive feedback that helped improve our manuscript. Please find below a point-by-point response to your comments:
- The authors wish to thank the reviewer for making us aware of these recently published manuscripts involving mesenchymal stromal cell derived extracellular vesicles in the therapy of experimental bronchopulmonary dysplasia. In the revised manuscript we include the publications of Lithopoulos et al. (2022), Ai et al. (2022) and Wu et al. (2021). We do not include the publication of Bellio et al. (2021) as our review only includes publications involving extracellular vesicles derived from mesenchymal stromal cells. The publication of Bellio et al. does not handle about mesenchymal stromal cells.
- The revised manuscript contains a paragraph on bio-engineering strategies to enhance the therapeutic potential of extracellular vesicles.
- We have screened the text and the tables for typographical errors.
Sincerely,
Dr. Bernard Thébaud
Dr. Flore Lesage